# Transition Landmarks from Abstraction Cuts

**Primary Keywords:** *None*

## Abstract

We introduce *transition-counting constraints* as a principled tool to formalize constraints that must hold in every solution of a transition system. We then show how to obtain *transition landmark* constraints from *abstraction cuts*. Transition landmarks dominate operator landmarks in theory but require solving a linear program that is prohibitively large in practice. We compare different constraints that project away transition-counting variables and then further relax the constraint. For one important special case, we provide a lossless projection. We finally discuss efficient data structures to derive cuts from abstractions and store them in a way that avoids repeated computation in every state. We compare the resulting heuristics both theoretically and on the IPC benchmarks.

## Introduction

Operator counting (Pommerening et al. 2014) is a framework in classical planning to combine information from different sources. It uses constraints on the number of times each operator of the task is used. One source of inaccuracy in this framework is that one planning operator usually induces many transitions. Information on how often an operator is used aggregates over these transitions an thus cannot distinguish different uses of the same operator.

We introduce transition counting as an extension of operator counting to a more fine grained level. Since the number of transitions in a planning task is prohibitively large, we show how to derive transition-counting constraints from abstractions. Such constraints are more compact and can be linked to the operator-counting framework as well.

In particular, we investigate transition landmarks based on cuts in a transition system. These can have an advantage over operator landmarks derived from the same information as they distinguish different uses of the same operator. We then explore what effect these constraints have on the operator-counting variables as this is the only way they can interact with other constraints. Projections of cut-based constraints to operator-counting variables can still be large but can be relaxed further, trading off accuracy for a smaller size.

We also introduce a way of finding cuts in transition systems based on LM-cut (Helmert and Domshlak 2009) and an efficient data structure to access them during a search.

Finally, we empirically evaluate the constraints and their relaxations on IPC benchmarks.

## Background

**Classical planning**  We consider *planning task* with operators $\mathcal{O}$ and a cost function $cost : \mathcal{O} \to \mathbb{R}_0$ (e.g., Bäckström and Nebel 1995; Haslum et al. 2019). Further details of the planning framework are not important as we work on the induced transition systems.

A planning task $\Pi$ induces a transition system $\mathcal{T}(\Pi) = \langle \mathcal{S}, \mathcal{O}, T, I, G, cost \rangle$ with a finite set of *states* $\mathcal{S}$; the *operators* $\mathcal{O}$ of $\Pi$; a finite set of *transitions* $T \subseteq \mathcal{S} \times \mathcal{O} \times \mathcal{S}$ where an individual transition is written as $s \xrightarrow{o} s'$ and $label(s \xrightarrow{o} s') = o$ is used to denote its *label*; an *initial state* $I \in \mathcal{S}$; a set of *goal states* $G \subseteq \mathcal{S}$; and the *transition cost function cost* $: T \to \mathbb{R}_0$ is defined based on the cost function of $\Pi$ as $cost(t) = cost(label(t))$.

A sequence of transitions $\pi = \langle t_1, \dots, t_n \rangle$ is an *s-plan* for $\mathcal{T}(\Pi)$ iff $t_i \in T$ with $t_i = s_{i-1} \xrightarrow{o_i} s_i$ for $1 \le i \le n$, $s_0 = s$ and $s_n \in G$. If $s = I$, $\pi$ is a *plan for* $\Pi$. The cost of an $s$-plan $\pi = \langle t_1, \dots, t_n \rangle$ is the sum over the transition costs of the sequence, i.e., $cost(\pi) = \sum_{i=1}^{n} cost(t_i)$. An $s$-plan is *optimal* if it has minimal cost among all $s$-plans. The *perfect heuristic* $h^\star$ maps each state $s$ to the cost of an optimal $s$-plan and to $\infty$ if no such plan exists. A function $h : \mathcal{S} \to \mathbb{R}_0^+ \cup \{\infty\}$ is called an *admissible heuristic* if $h \le h^\star$.

**Landmarks**  In this work we consider *disjunctive action landmarks* (Zhu and Givan 2003; Helmert and Domshlak 2009; Büchner, Keller, and Helmert 2021) and call them *landmarks* for brevity. Let $\mathcal{T}$ be a transition system with operators $\mathcal{O}$ and let $s$ be one of its states. We call a set of operators $\mathcal{L} \subseteq \mathcal{O}$ a *landmark* for $s$ if for all $s$-plans $\pi = \langle t_1, \dots, t_n \rangle$ there is an $1 \le i \le n$ such that $label(t_i) \in \mathcal{L}$.

**Abstractions**  An *abstraction function* $\alpha$ maps all states of a transition system $\mathcal{T}$ to a set of states $\mathcal{S}^\alpha$, and a *label reduction* $\lambda$ maps its operators to a set of *labels* $\mathcal{O}^\lambda$. Such abstraction functions induce an *abstract transition system* (or *abstraction*) $\mathcal{T}^{\alpha\lambda} = \langle \mathcal{S}^\alpha, \mathcal{O}^\lambda, T^{\alpha\lambda}, I^\alpha, G^\alpha, cost^\lambda \rangle$ where $T^{\alpha\lambda} = \{\alpha(s) \xrightarrow{\lambda(o)} \alpha(s') \mid s \xrightarrow{o} s' \in T\}$, $I^\alpha = \alpha(I)$, $G^\alpha = \{\alpha(s) \mid s \in G\}$, and $cost^\lambda(s \xrightarrow{\ell} s') \le \min_{o \in \lambda^{-1}(\ell)} cost(o)$, where $T$ are the transitions and $G$ the goal states of $\mathcal{T}(\Pi)$. Usually, $\alpha$ is chosen such that $\mathcal{T}^\alpha$ is much smaller than $\mathcal{T}(\Pi)$, e.g., as projecting $s$ to a small subset $P$ of variables. In that case the abstraction is called a *projection* (or *atomic projection* if $|P| = 1$). A common choice for $\lambda$ is *exact label reduction* that maps operators to

the same label iff they have the same cost and induce transitions between the same pairs of abstract states.

An important property of abstractions is that all paths in $\mathcal{T}$ are also feasible in $\mathcal{T}^{\alpha\lambda}$. Given a sequence of transitions $\langle s_0 \xrightarrow{o_1} s_1, \ldots, s_{n-1} \xrightarrow{o_n} s_n \rangle$ such that $s_{i-1} \xrightarrow{o_i} s_i \in T$ for all $1 \leq i \leq n$, it holds that $\alpha(s_{i-1}) \xrightarrow{\lambda(o_i)} \alpha(s_i) \in T^\alpha$. Put differently, the cost of an optimal plan in the abstract transition system is a lower bound on the cost of an optimal plan in the original transition system. The perfect heuristic for the abstract transition system ($h^{\alpha\lambda}$) is therefore admissible for the original transition system.

**Operator Counting**  Consider a set of abstractions $A = \{\mathcal{T}^{\alpha_1\lambda_1}, \ldots, \mathcal{T}^{\alpha_n\lambda_n}\}$. Each heuristic $h^{\alpha_i\lambda_i}$ provides an admissible estimate, but to use them in an optimal search algorithm like A$^\star$ we have to combine them in a way that maintains admissibility. *Operator counting* (Pommerening et al. 2014) is a framework to admissibly combine admissible heuristics that can be expressed in terms of necessary plan properties called *operator-counting constraints*.

We denote with $occur(o, \pi)$ the number of occurrences of operator $o \in \mathcal{O}$ in a plan $\pi$. A set of linear inequalities over a set of non-negative real-valued and integer variables $Y$ which includes an integer variable $Y_o$ for each $o \in \mathcal{O}$ (and any number of additional variables) is an *operator-counting constraint* for state $s$ if for all $s$-plans $\pi$ there exists a feasible solution with $Y_o = occur(o, \pi)$ for all $o \in \mathcal{O}$. The objective value of the *operator-counting integer/linear program* can be used as an admissible heuristic for planning. It is computed by minimizing the function $\sum_{o \in \mathcal{O}} cost(o) \cdot Y_o$ subject to a given set of operator-counting constraints.

Operator-counting constraints have been derived from a variety of planning heuristics, including landmarks and orderings between landmarks (e.g., Büchner, Keller, and Helmert 2021), the delete relaxation (e.g., Helmert and Domshlak 2009; Imai and Fukunaga 2014) and net change constraints (Bonet 2013). Abstraction heuristics have also been used before to derive operator-counting constraints. Pommerening, Röger, and Helmert (2013) introduce *post-hoc optimization constraints* which describe the relationship between operators that are relevant for the abstraction, their cost and the heuristic value of the abstraction, and Seipp, Keller, and Helmert (2021) strengthen post-hoc optimization constraints by taking saturated costs into account.

## Transition Counting

Pommerening et al. (2014) show that cost partitioning (Katz and Domshlak 2010) of abstraction heuristics can be encoded in the operator-counting framework. They use auxiliary variables for the number of times each abstract transition is used. This turns out to be unnecessary as the same heuristic can be encoded more compactly without these variables but it suggests an extension of the operator-counting framework to the level of transitions.

**Definition 1.** *Let $\langle \mathcal{S}, \mathcal{O}, T, I, G, cost \rangle$ be a transition system and let $s \in \mathcal{S}$. Let $Y$ be a set of non-negative real-valued and integer variables, including non-negative (integer-valued) operator- and transition-counting variables $Y_o$ for each $o \in$*

$\mathcal{O}$ *and $Y_t$ for each $t \in T$ along with any number of additional variables. We denote the number of occurrences of operators $o \in \mathcal{O}$ and transitions $t \in T$ in $\pi$ with $occur(o, \pi)$ and $occur(t, \pi)$. A set of linear inequalities over $Y$ is called a* transition-counting constraint *for $s$ if for all $s$-plans $\pi$ there exists a feasible solution with $Y_o = occur(o, \pi)$ for all $o \in \mathcal{O}$ and $Y_t = occur(t, \pi)$ for all $t \in T$.*

*A* transition constraint set *for $s$ is a set of transition-counting constraints for $s$ where the only common variables between constraints are the counting variables.*

By definition, every transition-counting constraint and every transition constraint set is an operator-counting constraint. (Note that we have to consider a set of transition-counting constraints as a single operator-counting constraint because they share variables other than $Y_o$.)

If we want transition-counting constraints to be useful in the operator-counting framework, we have to link the transition counts to the operator counts.

**Definition 2.** *Consider a transition system $\mathcal{T}$ with operators $\mathcal{O}$ and transitions $T$. The* linking constraint *is the set of linear inequalities $c^{\text{link}(\mathcal{T})}$:*

$$\sum_{\substack{t \in T \\ label(t)=o}} Y_t = Y_o \qquad \text{for all } o \in \mathcal{O}.$$

Please observe that the linking constraint is a transition-counting (and thus also an operator-counting) constraint as the equations obviously holds for any plan.

Using variables for every transition in a planning task $\Pi$ is typically intractable but we can do so for small abstractions of $\Pi$. Unless the label reduction is the identity, though, abstractions use labels different from $\mathcal{O}$. The following constraint translates between $\mathcal{O}$ and $\mathcal{O}^\lambda$.

**Definition 3.** *Let $\lambda$ be a label reduction. The* translation constraint for $\lambda$ *is the set of linear inequalities $c^{\text{translate}(\lambda)}$:*

$$\sum_{o \in \lambda^{-1}(\ell)} Y_o = Y_\ell \qquad \text{for all } \ell \in \mathcal{O}^\lambda.$$

With this constraint, we can use operator-counting constraints from abstractions.

**Proposition 1.** *Let $\mathcal{T}^{\alpha\lambda}$ be an abstraction of planning task $\Pi$ and let $c$ be an operator-counting constraint for a state $\alpha(s)$ of $\mathcal{T}^{\alpha\lambda}$. Then $\{c, c^{\text{translate}(\lambda)}\}$ is an operator-counting constraint for $s$ in $\Pi$.*

*Proof.* Let $\pi$ be an $s$-plan and let $\lambda(\pi)$ be the sequence that uses label $\lambda(o)$ where $\pi$ uses $o$. Since $\mathcal{T}^{\alpha\lambda}$ is an abstraction of $\mathcal{T}(\Pi)$, $\lambda(\pi)$ is an $\alpha(s)$-plan. As $c$ is an operator-counting constraint for $\alpha(s)$ in $\mathcal{T}^{\alpha\lambda}$, there is a solution of $c$ with $Y_\ell = occur(\ell, \lambda(\pi)) = \sum_{o \in \lambda^{-1}(\ell)} occur(o, \pi)$. Extending it with $Y_o = occur(o, \pi)$ also satisfies $c^{\text{translate}(\lambda)}$. $\square$

Note that $c^{\text{translate}(\lambda)}$ can be used to eliminate variables $Y_\ell$ from $c$ (replacing $Y_\ell$ with $\sum_{o \in \lambda^{-1}(\ell)} Y_o$) so the resulting constraint is in terms of the original operators.

We can thus derive transition-counting constraints for an abstraction, link them together with a linking constraint, and

then use Proposition 1 to get an operator-counting constraint for the original task. In the remainder of this paper, we thus usually ignore the fact that a transition system is an abstraction and just treat it as the only transition system.

## Comparing Operator-Counting Constraints

Since auxiliary variables like transition-counting variables are not shared between different operator-counting constraints, we compare their relative strength solely based on their solutions for operator-counting variables.

**Definition 4.** *Let $c$ be an operator-counting constraint with $n$ operator-counting variables $Y$ and $m$ auxiliary variables $Z$. The real-valued operator-counting solutions of $c$ are $\mathrm{Sol}_{\mathrm{LP}}(c) := \{Y \in \mathbb{R}^n \mid \exists Z \in \mathbb{R}^m \text{ where } Y, Z \text{ satisfies } c\}$.*

We say an operator-counting constraint $c_1$ is *implied* by a constraint $c_2$ for real-valued variables if $\mathrm{Sol}_{\mathrm{LP}}(c_2) \subseteq \mathrm{Sol}_{\mathrm{LP}}(c_1)$. If there are solutions in $\mathrm{Sol}_{\mathrm{LP}}(c_1)$ that are not in $\mathrm{Sol}_{\mathrm{LP}}(c_2)$, we say that $c_1$ is *weaker* than $c_2$ (or that $c_2$ is *stronger*) wrt. real-valued variables. If $\mathrm{Sol}_{\mathrm{LP}}(c_2) = \mathrm{Sol}_{\mathrm{LP}}(c_1)$, both constraints imply each other, and we say they are *equivalent* wrt. real-valued variables. Using a stronger operator-counting constraint in an operator-counting LP can only increase the objective value, while replacing a constraint with an equivalent one will not change the objective value, even in the presence of other constraints. A way to show that $c_1$ is implied by $c_2$ for real-valued variables is to show that the inequalities in $c_1$ are conic combinations of the inequalities in $c_2$. For example, $c_2 := \{2Y_t - Y_a \geq 0, Y_b - 2Y_t \geq 0\}$ implies $c_1 := \{Y_b \geq Y_a\}$ because the inequality is the sum of the inequalities in $c_2$.

Analogously to the definitions above we define the set of integer-valued solutions $\mathrm{Sol}_{\mathrm{IP}}(c)$ and the terms *implied*, *weaker*, *stronger*, and *equivalent* wrt. integer-valued variables. We know that $\mathrm{Sol}_{\mathrm{IP}}(c) \subseteq \mathrm{Sol}_{\mathrm{LP}}(c) \cap \mathbb{N}^n$ but not all integer-valued solutions in $\mathrm{Sol}_{\mathrm{LP}}(c)$ are necessarily in $\mathrm{Sol}_{\mathrm{IP}}(c)$. Consider for example a constraint $c := \{Y_a \leq 2Y_t, 2Y_t \leq Y_b\}$. We have $\mathrm{Sol}_{\mathrm{LP}}(c) = \{Y \mid Y_a \leq Y_b\}$, and $Y_a = Y_b = 1$ is a solution in $\mathrm{Sol}_{\mathrm{LP}}(c) \cap \mathbb{N}^n$. However, this solution is not in $\mathrm{Sol}_{\mathrm{IP}}(c)$, as there is no integer $Y_t \in \mathbb{N}$ with $Y_a = 1 \leq 2Y_t \leq 1 = Y_b$.

An important result from Operations Research (e.g., Conforti, Cornuéjols, and Zambelli 2014) shows that if the coefficient matrix of the auxiliary variables is totally unimodular, then integer-valued solutions in $\mathrm{Sol}_{\mathrm{LP}}(c) \cap \mathbb{N}^n$ can always be extended to an integer-valued solution of $c$, so $\mathrm{Sol}_{\mathrm{LP}}(c) \cap \mathbb{N}^n = \mathrm{Sol}_{\mathrm{IP}}(c)$. In that case, if a constraint $c_1$ is implied by $c_2$ wrt. real-valued variables, it is also implied wrt. integer variables, and if it is weaker wrt. integer variables, it is also weaker wrt. real-valued variables. In such cases we just use the terms *implied* and *weaker* without reference to integer or real-valued variables.

## Cut Landmarks

We now focus on specific transition-counting constraints related to landmarks. So far, landmarks are derived with label propagation in the relaxed task graph (Zhu and Givan 2003; Keyder, Richter, and Helmert 2010), backward-chaining in the delete relaxation (Hoffmann, Porteous, and

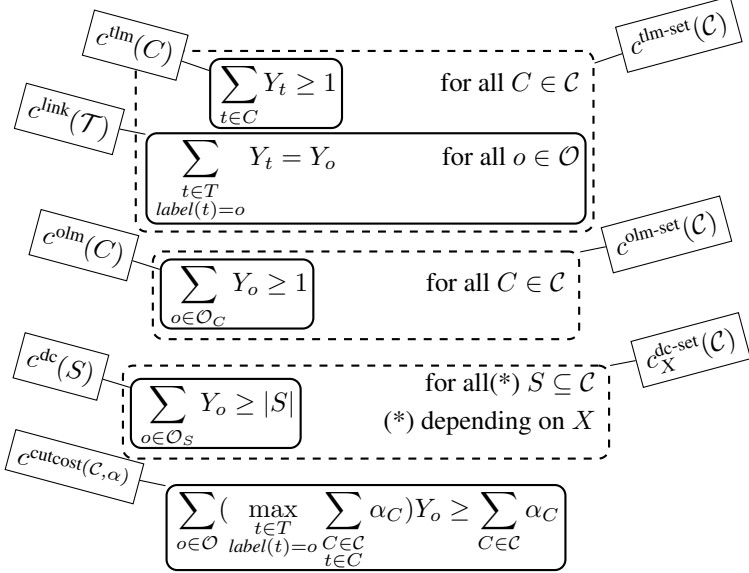

Figure 1: Cut-based constraints introduced in the paper.

Sebastia 2004; Richter, Helmert, and Westphal 2008), or as cuts in a graph justifying cost of a relaxed task (Helmert and Domshlak 2009). We propose a new method of deriving landmarks from cuts in (abstract) transition systems.

**Definition 5.** *Let $\mathcal{T} = \langle \mathcal{S}, \mathcal{O}, T, I, G, cost \rangle$ be a transition system and $s$ a state in $\mathcal{S}$. A set of transitions $C \subseteq T$ is an $s$-cut iff there is a subset of states $Z \subset \mathcal{S}$ with $s \notin Z$, $G \subseteq Z$, and $C = \{s_1 \xrightarrow{o} s_2 \in T \mid s_1 \notin Z, s_2 \in Z\}$.*

Such cuts induce disjunctive action landmarks that must hold in every plan for the transition system.

**Proposition 2.** *If $C$ is an $s$-cut then the set of operators $\mathcal{O}_C := \{label(t) \mid t \in C\}$ is a landmark for $s$.*

In this and the following sections we introduce several constraints based on cuts. Figure 1 collects the most important ones in a single place and is meant for quick reference. The first such constraint uses transition-counting constraints to be more fine-grained than just talking about $\mathcal{O}_C$.

**Proposition 3.** *If $C$ is an $s$-cut then the* transition landmark constraint $c^{\mathrm{tlm}(C)}$:

$$\sum_{t \in C} Y_t \geq 1$$

*is a transition-counting constraint.*

If the transition landmark constraint for a single cut $C$ is linked to operator counts, it contributes the same information to the operator-counting framework as the landmark $\mathcal{O}_C$. (We defer proving this statement to Proposition 7, where we prove it in a more general form.)

**Proposition 4.** *Let $C$ be an $s$-cut derived from a transition system $\mathcal{T}$. Then the* operator landmark constraint $c^{\mathrm{olm}(C)}$:

$$\sum_{o \in \mathcal{O}_C} Y_o \geq 1$$

*is equivalent to $\{c^{\mathrm{tlm}(C)}, c^{\mathrm{link}(\mathcal{T})}\}$.*

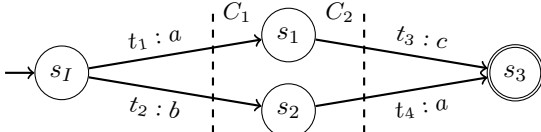

Figure 2: Example transition system with two cuts $C_1$ and $C_2$. Transition labels give transition and operator, separated by a colon.

The advantage of transition landmarks over operator landmarks becomes clear if we use more than one cut. Consider the transition system and the two $s_I$-cuts $C_1$ and $C_2$ depicted in Figure 2. Operator landmark constraints just use the disjunctive action landmarks derived from these cuts:

$$Y_a + Y_b \geq 1 \qquad c^{\mathrm{olm}(C_1)}$$
$$Y_a + Y_c \geq 1 \qquad c^{\mathrm{olm}(C_2)}$$

These constraints have a solution $Y_a = 1$ and $Y_b = Y_c = 0$. However, we can see in the transition system that using operator $a$ once will not suffice to solve the task. If we distinguish transitions in the constraints, two cuts with different transitions that are labeled with the same operator can no longer be resolved with just a single use of the operator. The transition landmark constraints for the cuts are

$$Y_{t_1} + Y_{t_3} \geq 1 \qquad c^{\mathrm{tlm}(C_1)}$$
$$Y_{t_2} + Y_{t_4} \geq 1 \qquad c^{\mathrm{tlm}(C_2)}$$

Together with the linking constraint $c^{\mathrm{link}(\mathcal{T})}$, in particular the equation $Y_{t_1} + Y_{t_2} = Y_a$, the assignment from above is no longer a solution. In fact, every solution requires a total operator count of at least 2.

**Definition 6.** *Let $\mathcal{T}$ be a transition system, $s$ one of its states, and $\mathcal{C}$ a set of $s$-cuts in $\mathcal{T}$. The* transition landmark set constraint *for $\mathcal{C}$ is the transition constraint set $c^{\mathrm{tlm\text{-}set}(\mathcal{C})} := \{c^{\mathrm{tlm}(C)} \mid C \in \mathcal{C}\} \cup \{c^{\mathrm{link}(\mathcal{T})}\}$.*

We get the following connection to operator landmarks as a corollary of Proposition 4 and the example above.

**Proposition 5.** *Let $\mathcal{C}$ be a set of $s$-cuts derived from a transition system $\mathcal{T}$. Then $c^{\mathrm{tlm\text{-}set}(\mathcal{C})}$ implies $c^{\mathrm{olm\text{-}set}(\mathcal{C})} := \{c^{\mathrm{olm}(\mathcal{O}_C)} \mid C \in \mathcal{C}\}$ and can be stronger.*

Even though the transition landmark set constraint dominates the operator landmark set constraint in terms of heuristic guidance, it comes with a significant drawback: it requires many auxiliary variables and linking constraints which, as we will see in our experimental evaluation, impair planner performance in practice. We are therefore interested in projecting out transition-counting variables or approximating the projection.

## Disjoint Cuts Constraints

If cuts are disjoint, this means every transition occurs in at most one cut. It is important to note that this only refers to the transitions. An operator can still be used to label different transitions occurring in multiple cuts. In our running example, cuts $C_1$ and $C_2$ are disjoint because they do not share a transition, even though both cuts mention operator $a$. In fact, if the induced landmarks of the cut are disjoint as well, transition landmark set constraints have no advantage over operator landmark set constraints.

If we consider a set of cuts $S$ (say $S = \{C_1, C_2\}$ in our running example), every plan has to pass through all of the cuts in $S$. If cuts in $S$ are pairwise disjoint, this takes $|S|$ separate transitions. These transitions can only be labeled with operators in $\mathcal{O}_S := \{label(t) \mid t \in C, C \in S\} = \bigcup_{C \in S} \mathcal{O}_C$, i.e., the operators mentioned anywhere in $S$. (In our example, $\mathcal{O}_S = \{a, b, c\}$.) We can conclude that the sum of operator-counts for operators in $\mathcal{O}_S$ must be at least $|S|$. In the example, this corresponds to the constraint $Y_a + Y_b + Y_c \geq 2$. This argument holds in general, so the following constraints are operator-counting constraints:

**Definition 7.** *Let $S$ be a set of pairwise disjoint $s$-cuts. The* disjoint cuts constraint *for $S$ is $c^{\mathrm{dc}(S)}$:*

$$\sum_{o \in \mathcal{O}_S} Y_o \geq |S|.$$

To see the importance of cuts being pairwise disjoint, consider that we add the cut $C_3 = \{t_2, t_3\}$ to the example. The disjoint cut constraint for $S = \{C_1, C_2, C_3\}$ would exclude plans of length 2, which clearly exist.

When comparing disjoint cuts constraints to operator landmark constraints, we first observe that $c^{\mathrm{olm}(C)}$ for a cut $C$ is the same as the $c^{\mathrm{dc}(S)}$ for a set of cuts $S = \{C\}$. For larger sets $S$, the constraint becomes incomparable to operator landmark constraints. We can see this in our running example, where $\{c^{\mathrm{olm}(C_1)}, c^{\mathrm{olm}(C_2)}\} = \{Y_a + Y_b \geq 1, Y_a + Y_c \geq 1\}$ and $\{c^{\mathrm{dc}(\{C_1, C_2\})}\} = \{Y_a + Y_b + Y_c \geq 2\}$. The assignment $Y_a = Y_c = 0$ and $Y_b = 2$ satisfies $Y_a + Y_b + Y_c \geq 2$ but not $Y_a + Y_c \geq 1$, while the assignment $Y_a = 1$ and $Y_b = Y_c = 0$ satisfies $Y_a + Y_b \geq 1$ and $Y_a + Y_c \geq 1$ but not $Y_a + Y_b + Y_c \geq 2$.

We now show that multiple disjoint cuts constraints can be used to express a constraint equivalent to $c^{\mathrm{tlm\text{-}set}(\mathcal{C})}$ if cuts in $\mathcal{C}$ are pairwise disjoint. We start by showing that $c^{\mathrm{dc}(S)}$ is implied by $c^{\mathrm{tlm\text{-}set}(\mathcal{C})}$ for every subset of cuts $S \subseteq \mathcal{C}$.

**Proposition 6.** *Let $\mathcal{C}$ be a set of pairwise disjoint $s$-cuts and $S \subseteq \mathcal{C}$ one of its subsets. Then $c^{\mathrm{dc}(S)}$ is implied by $c^{\mathrm{tlm\text{-}set}(\mathcal{C})}$ but can be weaker, even for $S = \mathcal{C}$.*

*Proof.* To show that $c^{\mathrm{dc}(S)}$ is implied, we can sum up constraints $c^{\mathrm{tlm}(C)}$ for $C \in S$. On the right-hand side, this adds 1 for each $C \in S$, resulting in $|S|$. Sums for different cuts do not overlap, so each transition-counting variable occurs at most once on the left-hand side. We can then add constraint $Y_t \geq 0$ to include transition-counting variables on the left-hand side that belong to operators in $\mathcal{O}_S$ but do not occur in the sum yet. Afterwards, we can use equations from $c^{\mathrm{link}(\mathcal{T})}$ for all $o \in \mathcal{O}_S$ to replace sums of transition-counting variables with the corresponding operator-counting variables.

To see that $c^{\mathrm{dc}(S)}$ for $S = \mathcal{C}$ can be weaker than $c^{\mathrm{tlm\text{-}set}(\mathcal{C})}$ consider our running example: $Y_a = Y_c = 0$ and $Y_b = 2$ is a solution for $c^{\mathrm{dc}(\mathcal{C})}$ but not for $c^{\mathrm{tlm\text{-}set}(\mathcal{C})}$. $\square$

While a single disjunctive cuts constraint can be weaker than the transition landmark set constraint, including such constraints for all subsets $S \subseteq \mathcal{C}$ makes the resulting constraint equivalent:

**Proposition 7.** *Let $\mathcal{C}$ be a set of pairwise disjoint s-cuts in a transition system $\mathcal{T}$. Then $c^{\text{tlm-set}(\mathcal{C})}$ and $c_{\text{all}}^{\text{dc-set}(\mathcal{C})} := \{c^{\text{dc}(S)} \mid S \subseteq \mathcal{C}\}$ are equivalent.*

We refer to the technical report (Anonymized 2023) for the full derivation and only present a rough sketch here.

*Proof sketch.* The coefficient matrix of $c^{\text{tlm-set}(\mathcal{C})}$ is totally unimodular if cuts are pairwise disjoint, so showing equivalence wrt. real-valued variables is sufficient. For this case, we consider combinations of constraints $c^{\text{tlm}(C)}$, $c^{\text{link}(\mathcal{T})}$, and $Y_t \geq 0$, similar to the approach in the proof of Proposition 6. However, instead of considering one specific combination, we consider *all* conic combinations by multiplying the coefficient matrix with the extreme rays of a cone $D$ that describes conditions for all $Y_t$ reaching a coefficient of $0$. For a set of multipliers $\alpha$, the resulting combination of inequalities is the constraint $c^{\text{cutcost}(\mathcal{C}, \alpha)}$ :

$$\sum_{o \in \mathcal{O}} (\max_{\substack{t \in T \\ label(t)=o}} \sum_{\substack{C \in \mathcal{C} \\ t \in C}} \alpha_C) Y_o \geq \sum_{C \in \mathcal{C}} \alpha_C$$

and $c^{\text{tlm-set}(\mathcal{C})}$ is equivalent to $\{c^{\text{cutcost}(\mathcal{C}, \alpha)} \mid \alpha \geq 0\}$.

We show that for disjoint cuts, a vector $\alpha$ can only be an extreme ray of the cone $D$ if it is binary, so $c^{\text{tlm-set}(\mathcal{C})}$ is equivalent to $\{c^{\text{cutcost}(\mathcal{C}, \alpha)} \mid \alpha \in \{0,1\}^{|\mathcal{C}|}\}$. Each binary vector $\alpha$ corresponds to a subset $S_\alpha \subseteq \mathcal{C}$ with $S_\alpha = \{C \in \mathcal{C} \mid \alpha_C = 1\}$. We can then see that

$$\max_{\substack{t \in T \\ label(t)=o}} \sum_{\substack{C \in \mathcal{C} \\ t \in C}} \alpha_C = \begin{cases} 1 & o \in \mathcal{O}_{S_\alpha} \\ 0 & \text{otherwise} \end{cases} \quad \text{and} \quad \sum_{C \in \mathcal{C}} \alpha_C = |S_\alpha|.$$

This shows that $c^{\text{cutcost}(\mathcal{C}, \alpha)}$ exactly matches $c^{\text{dc}(S_\alpha)}$. Thus $\{c^{\text{cutcost}(\mathcal{C}, \alpha)} \mid \alpha \in \{0,1\}^{|\mathcal{C}|}\}$ is equivalent to $c^{\text{dc-set}(\mathcal{C})}$. $\square$

Say two cuts $C_1, C_2$ are *share an operator* if $\mathcal{O}_{C_1} \cap \mathcal{O}_{C_2} \neq \emptyset$, i.e, some operator is used as a label in both of the cuts. For a set of cuts $S$, consider the graph $G_S$ where each $C \in S$ is a node and there is an edge between two cuts iff they share an operator. We say $S$ is *connected* if $G_S$ has a single connected component. We now show that we only have to consider connected subsets $S \subseteq \mathcal{C}$. If $S$ is not connected, consider a connected component $S_1$ of $G_S$ and $S_2 = S \setminus S_1$. In that case, $\mathcal{O}_{S_1}$ and $\mathcal{O}_{S_2}$ are disjoint, so $c^{\text{dc}(S_1)} + c^{\text{dc}(S_2)} = c^{\text{dc}(S)}$, i.e., $c^{\text{dc}(S)}$ is implied by the other two constraints. Together with Proposition 7 we get the following result:

**Theorem 1.** *Let $\mathcal{C}$ be a set of pairwise disjoint s-cuts in a transition system $\mathcal{T}$. Then $c^{\text{tlm}(\mathcal{C})}$ and $c_{\text{connected}}^{\text{dc-set}(\mathcal{C})} := \{c^{\text{dc}(S)} \mid S \subseteq \mathcal{C}, S \text{ is connected}\}$ are equivalent.*

While $c^{\text{tlm}(\mathcal{C})}$ and $c_{\text{connected}}^{\text{dc-set}(\mathcal{C})}$ express the same restriction on the operator-counting variables they differ in size. In a transition system with transitions $T$, and with a set of

cuts $\mathcal{C}$, $c^{\text{tlm}(\mathcal{C})}$ has $|\mathcal{C}| + |\mathcal{O}|$ inequalities and $|\mathcal{O}| + |T|$ variables. The equivalent disjoint cuts set constraint has $O(2^{|\mathcal{C}|})$ inequalities but only $|\mathcal{O}|$ variables. In transition systems with a large number of transitions, this can pay off. Additionally, the constraint can easily be relaxed by dropping some of the disjunctive cuts constraints to get a weaker but more compact representation. In one extreme, only atomic subsets ($c_{\text{atomic}}^{\text{dc-set}(\mathcal{C})} := \{c^{\text{dc}(S)} \mid S \subseteq \mathcal{C}, |S| = 1\}$) are considered and the constraints relaxes to $c^{\text{olm-set}(\mathcal{C})}$. In the other extreme all connected subsets are considered and the constraint is equivalent to $c^{\text{tlm}(\mathcal{C})}$. But there are options in between, for example $c_{\text{atomic,max}}^{\text{dc-set}(\mathcal{C})} := \{c^{\text{dc}(S)} \mid S \subseteq \mathcal{C}, |S| = 1 \text{ or } S \text{ is a maximally connected subset of } \mathcal{C}\}$. This constraint has at most $2|\mathcal{C}|$ constraints and dominates the operator landmark set constraint.

## Overlapping Cuts Constraints

To get a better understanding of $c^{\text{tlm-set}(\mathcal{C})}$, let us first consider how a transition-counting LP combines the constraints $c^{\text{tlm}(C)}$ for $C \in \mathcal{C}$. A known result for operator counting states that combining operator-counting constraints in an operator-counting LP is equivalent to optimal operator cost partitioning over heuristics that each consider one constraint individually (Pommerening et al. 2015). This result generalizes to transition cost partitioning (Keller et al. 2016) in a straightforward way.

Let $\sum_{C \in \mathcal{C}} cost_C \leq cost$ be the transition cost partition computed by this combination. Assigning different costs to two transitions $t_1, t_2 \in C$ can never be useful, as the cheapest way of satisfying a single cut is to use its cheapest transition. Assigning negative costs to a transition also cannot be useful as there is no upper limit how often a transition within a cut can be used. Assigning negative costs would make the heuristic value for this cut arbitrarily low, so it cannot be part of an optimal cost partition. Finally, assigning non-zero costs $cost_C(t) > 0$ for some transition $t \notin C$ can never be beneficial. We can thus limit attention to cost functions $cost_C$ defined as

$$cost_C(t) = \begin{cases} \alpha_C & \text{if } t \in C \\ 0 & \text{otherwise} \end{cases}$$

for some $\alpha_C \geq 0$.

The operator-counting constraint $c^{\text{tlm-set}(\mathcal{C})}$ also contains the linking constraint in addition to the constraints discussed above. In the context of cost partitioning, it has the effect of restricting the transition cost functions to the operator cost function of the planning task. In our case, this requires that

$$cost(o) \geq \max_{\substack{t \in T \\ label(t)=o}} cost(t)$$

$$\geq \max_{\substack{t \in T \\ label(t)=o}} \sum_{C \in \mathcal{C}} cost_C(t) = \max_{\substack{t \in T \\ label(t)=o}} \sum_{\substack{C \in \mathcal{C} \\ t \in C}} \alpha_c$$

In the proof of Proposition 7 we saw that $c^{\text{tlm-set}(\mathcal{C})}$ is equivalent to $\{c^{\text{cutcost}(\mathcal{C}, \alpha)} \mid \alpha \geq 0\}$. This is true even for overlapping cuts and the argument above gives an intuition

on why this is the case: vectors $\alpha$ correspond to non-negative transition cost partitions over the cuts, and the coefficients of variables $Y_o$ correspond to cost functions that are just high enough to guarantee admissibility.

## Finding Cuts in Transition Systems

So far, we assumed a set of cuts $\mathcal{C}$ was given. We now show one method to extract $\mathcal{C}$ from a transition system. The intention is to use sufficiently small abstractions but the method works on any transition system.

The number of all possible cuts grows exponentially in the number of states, which quickly becomes prohibitive even in small transition systems, so we focus on generating informative cuts that cover different parts of the transition system. Cuts having large overlaps are more likely to be satisfied with a small set of transitions. Likewise, cuts that contain both cheap and expensive transitions are more likely to be satisfied by a cheap transition, wasting the cost of the more expensive ones. We thus prefer cuts that only overlap in expensive transitions.

Our method is an adaption of LM-Cut (Helmert and Domshlak 2009), where cuts in a justification graph are found as the boundary to an incrementally increasing goal zone. One difference is that in LM-Cut the justification graph can change from one iteration to the next, whereas in our case the transition system remains stable. Also, our algorithm tracks costs of individual transitions where LM-cut uses operator cost functions.

Given a transition system $\mathcal{T} = \langle \mathcal{S}, \mathcal{O}, T, I, G, cost \rangle$, we maintain a *goal zone* $Z \subseteq \mathcal{S}$ and a cost function *rem* that tracks the *remaining costs* of all transitions. These are initialized to $Z = G$ and $rem = cost$. We then generate cuts by iterating the following loop until termination:

1. Add states to $Z$ from which any state in $Z$ can be reached on a 0-cost path under cost function *rem*.
2. Compute the cut $C = \{s \xrightarrow{o} s' \in T \mid s \notin Z, s' \in Z\}$.
3. Terminate if $C = \emptyset$.
4. Reduce the remaining costs of each transition $t \in C$. We consider two variants here:
   - *disjoint variant:* Set $rem(t) = 0$.
   - *overlapping variant:* Decrease $rem(t)$ by the cost of the cheapest transition in $C$, i.e., by $\min_{t' \in C} rem(t')$.

The top of Figure 3 shows an example of the generated goal zones and cuts. In every iteration, the remaining cost of at least one transition is updated from a positive value to 0 and the source of that transition will be added to $Z$ in the next iteration. This can only happen at most $|\mathcal{S}|$ times, so the loop is guaranteed to terminate.

The disjoint variant adds all source states of the transitions in the cut to the goal zone in the next iteration. Consequently, a transition can occur in at most one cut and the resulting cuts are pairwise disjoint. In the overlapping variant, expensive transitions can occur in multiple cuts.

Let $Z_i$ be the goal zone in iteration $i$, $C_i$ be the generated cut, and $S_i = \{s \mid s \xrightarrow{o} s' \in C_i\}$. It is easy to see that $C_i$ is an $s$-cut for all states $s \in \mathcal{S} \setminus Z_i$. This includes the set $S_i$ but interestingly it also includes $S_j$ for all $j > i$. We can use

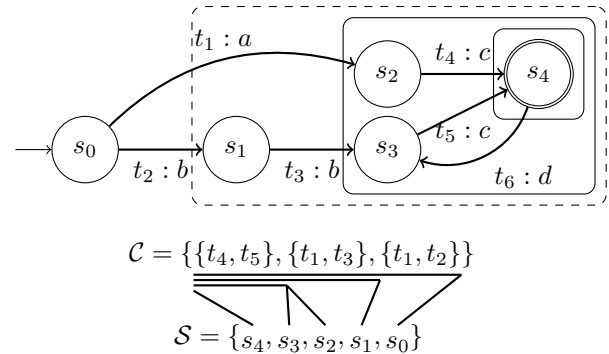

Figure 3: Transition system with $cost(t_1) = 2$ and $cost(t_i) = 1$ for $i > 1$. Boxes around states show the goal zone after each iteration. Transitions entering a goal zone are part of the cut. In the disjoint variant, the dashed cut is not found. Below is the set $\mathcal{C}$ of discovered cuts and our data structure mapping states $s \in \mathcal{S}$ to sets of $s$-cuts.

this connection to efficiently store the cuts: We store all cuts in a list $[C_1, \ldots, C_n]$ in the order they were created. We then store an index $i_s$ into this list for every state $s$, where $i_s$ is the last iteration in which $s \in S_i$. The bottom of Figure 3 has an example of this data structure. If we want to retrieve the set of $s$-cuts for some state $s$, this is just the first $i_s$ entries of the list. The index is zero for all states $s_g$ that are in the goal zone initially: none of the computed cuts are $s_g$-cuts for such state. Dead-end states $s_{de}$ that have no path to the goal use index $n$. Technically, all cuts are $s_{de}$-cuts for such states because the fact that every $s_{de}$-plan passes through a cut is vacuously true for all cuts if there are no $s_{de}$-plans.

When considering disjoint cuts constraints, we are also interested in subsets of cuts $S \subseteq \mathcal{C}$ where $\mathcal{C}$ is a set of $s$-cuts. Different states $s$ use different sets $\mathcal{C}$ and thus have different choices for $S$. However, as before, we can create a list of all possible choices of $S$ ordered in a way that all subsets involving only $C_1, \ldots C_i$ come before the first subset involving $C_{i+1}$. We can then quickly identify the relevant subsets $S$ for each state $s$. This is useful as we can set up a single linear program in which we only have to enable/disable constraints when switching from one state to another.

Even if not all subsets should be considered, this approach can be helpful. For example, consider the case, where we want to consider only maximally connected subsets $S \subset \mathcal{C}$. We can order the maximally connected subsets involving only cuts $C_1, \ldots C_i$ before any set involving $C_{i+1}$ again. States in $S_i$ have a maximally connected component $X_i$ that includes $C_i$. If $C_{i+1}$ shares an operator with $C_i$, then states in $S_{i+1}$ will contain a maximally connected component $\{C_{i+1}\} \cup X_i$. This is a superset of $X_i$, so $X_i$ is not maximally connected for states in $S_{i+1}$. Collecting the prefix of the list up to the entry for $S_{i+1}$ will thus collect more sets than just the maximally connected ones. However, this collection is still limited and supports efficient incremental computation, so we also use it in our experiments. We call it $c_{\overline{\max}}^{dcset(\mathcal{C})}$ and analogously define $c_{atomic,\overline{\max}}^{dcset(\mathcal{C})}$.

Our algorithm not only generates cuts but also partitions transition costs between them. Let $\alpha_C = \min_{t \in C} rem(t)$ be the minimal cost of a transition in the cut at the time it is discovered. For these values of $\alpha$ the functions $cost_C$ as defined in the previous section form a cost partitioning of the original cost function.

This is particularly interesting in the overlapping variant of the algorithm, where cuts are created according to $h^*$-layers in the transition system: The first cut is induced by a goal zone $Z_1$ consisting of all states $s$ with $h^*(s) = 0$. Then in each step the cut $C_i$ separates states $s \in Z_i$ with $h^*(s) < \sum_{j \le i} \alpha_{C_j}$ from states with a higher heuristic value. States $s$ in $S_i$ have exactly the goal distance $h^*(s) = \sum_{j \le i} \alpha_{C_j}$. Consider $c^{\text{cutcost}(\mathcal{C}, \alpha)}$ for these values of $\alpha$ in a state $s \in S_i$. The right-hand side simplifies to $\sum_{j \le i} \alpha_{C_j} = h^*(s)$. For each transition $t = s \xrightarrow{o} s'$ on the left-hand side, $\sum_{\substack{C \in \mathcal{C} \\ t \in C}} \alpha_C = h^*(s) - h^*(s')$. The cost function expressed in the coefficients on the left-hand side of $c^{\text{cutcost}(\mathcal{C}, \alpha)}$ can then be interpreted as $cost^{\text{sat}}(o) = \max_{s \xrightarrow{o} s' \in T}(h^*(s) - h^*(s'))$, the saturated cost function in the transition system. Constraint $c^{\text{cutcost}(\mathcal{C}, \alpha)}$ for this choice of $\alpha$ thus is the saturated post-hoc optimization constraint (Pommerening, Röger, and Helmert 2013): $\sum_{o \in \mathcal{O}} cost^{\text{sat}}(o) Y_o \ge h^*(s)$.

The above result implies that for the right set of cuts, $c^{\text{tlm-set}(\mathcal{C})}$ dominates saturated post-hoc optimization. We can show with an example that this dominance can be strict: In Figure 3, the saturated post-hoc optimization constraint is $2Y_a + Y_b + Y_c \ge 3$. It has a solution at $Y_c = 3, Y_a = Y_b = 0$ which does not satisfy the transition landmark $Y_c \ge 1$.

Many other ways of discovering cuts are possible. For example, one could also start from the initial state and generate landmarks in a forward direction. Another possible improvement would be to consider a *beyond-goal zone* as in LM-cut, of states that can only be reached after reaching a goal state. The graph could also be limited to states reachable from $s$ when computing landmarks for $s$. This can be useful for example in state $s_1$ in Figure 3 where $\{t_3\}$ is an $s_1$-cut but both variants of our method only find $\{t_1, t_3\}$. Using this cut would make efficient storage and re-use of the linear program during search more difficult though. We leave the exploration of other cut generation methods for future work and focus on the two variants described above.

## Connections to other Abstraction Heuristics

Figure 4 shows how the different constraints introduced here are related. For sets of pairwise disjoint cuts $\mathcal{C}$, the constraints $c_X^{\text{dc-set}(\mathcal{C})}$ generally get stronger, the more subsets $S \subseteq \mathcal{C}$ restriction $X$ considers. We have shown that if all (connected) subsets are used, the constraints can get as strong as $c^{\text{tlm-set}(\mathcal{C})}$ and reduce to $c^{\text{olm-set}(\mathcal{C})}$ when only atomic subsets are used. We also have shown examples where $c_{\text{atomic}}^{\text{dc-set}(\mathcal{C})}$ and $c_{\text{max}}^{\text{dc-set}(\mathcal{C})}$ are incomparable. All cut-based constraints are implied by $c^{\text{tlm-set}(\mathcal{C})}$ which can also be used for overlapping cuts in contrast to $c^{\text{dc-set}(\mathcal{C})}$.

So far these arguments can all be made with a single abstraction. As our overall aim is to extract constraints from

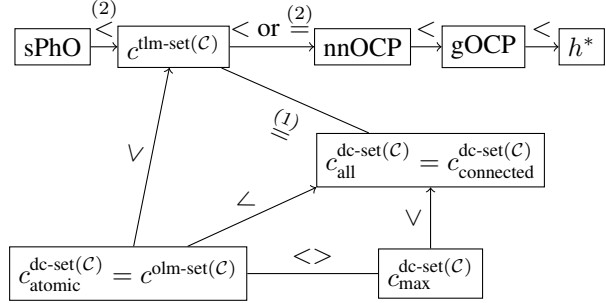

Figure 4: Relation of different cut-based constraints to each other and to other heuristics based on cost-partitioned abstractions. An edge marked $<$ from $c$ to $c'$ denotes that $c$ is is implied by $c'$ and sometimes weaker. Edges marked $=$ are between constraints that are equivalent wrt. operator-counting variables. The constraints marked $<>$ are incomparable. Limitations: (1) only for disjoint cuts, (2) only for sufficiently many cuts.

many small abstractions and combine them in operator-counting LP/IPs, we also compare this to other heuristics that do so. In the previous section, we have seen that $c^{\text{tlm-set}(\mathcal{C})}$ implies the saturated post-hoc optimization constraint if $\mathcal{C}$ contains sufficiently many cuts.

Finally, we try to find an upper bound on the heuristic quality and show that it can be tight. Combining constraints from different cuts in an operator-counting LP corresponds to optimal general cost partitioning. We have already seen that negative costs cannot contribute to our cut-based constraints. Since our constraints are operator-counting constraints, using all constraints from a single abstraction in a single operator-counting LP gives an admissible estimate for that abstraction. This value cannot exceed the cost of a shortest path in the abstraction. Non-negative cost partitioning of abstraction heuristics (called nnOCP in Figure 4) combines the cost of shortest paths in each abstraction, so our heuristics are limited by it.

For sufficiently many cuts, we can show that the operator-counting LP for $c^{\text{tlm-set}(\mathcal{C})}$ is optimal for its transition system under all non-negative cost functions: Consider a non-negative cost function *cost* and the set $\mathcal{C}$ of all $s$-cuts. Then there is one set $S \subseteq \mathcal{C}$ that separates heuristic layers according to *cost*. Any solution to the transition landmark constraints must pass through all those cuts and thus cause at least cost $h^*$, i.e., $\min\{\sum_{o \in \mathcal{O}} cost(o) Y_o \mid c^{\text{tlm-set}(\mathcal{C})}\} \ge h^*(s, cost)$. This means that if we combine constraints $c^{\text{tlm-set}(\mathcal{C})}$ for several abstractions in one operator-counting LP and include sufficiently many cuts in all cases, the resulting heuristic matches nnOCP over those abstractions.

## Experimental Evaluation

We implemented transition counting and our cut generation method for en Fast Downward version 22.06 (Helmert 2006). We use its operator-counting heuristic with CPLEX 22.1.1 as the LP solver to evaluate operator-counting LPs for

| | $c^{\text{tlm-set(disj)}}$ | $c^{\text{tlm-set(ovlp)}}$ | $c_{\text{all}}^{\text{dc-set(disj)}}$ | $c_{\text{connected}}^{\text{dc-set(disj)}}$ | $c_{\overline{\text{max}}}^{\text{dcset(disj)}}$ | $c_{\text{atomic}}^{\text{dc-set(disj)}}$ | $c_{\text{atomic},\overline{\text{max}}}^{\text{dc-set(disj)}}$ |
|---|---|---|---|---|---|---|---|
| Coverage | 732 | 755 | 743 | 783 | **807** | 805 | 803 |

Table 1: Number of solved tasks with different constraints.

our various constraints on the 1827 problems from the optimal tracks of the international planning competitions 1998–2018. Experiments are conducted on Intel Xeon Silver 4114 processors running on 2.2 GHz with a time limit of 30 minutes and a memory limit of 3.5 GiB. We will publish our code, data and benchmarks upon acceptance.

In this section, we write disj and ovlp for cuts discovered with the disjoint and overlapping variant of our cut generation algorithm. We use transition systems induced by the projections to all interesting patterns up to size 2 with exact label reduction (Pommerening, Röger, and Helmert 2013). We found that they yields best results among projection-based pattern generators implemented in Fast Downward.

Our baseline is $c^{\text{tlm-set(disj)}}$ which represents the full information of a given set of disjoint cuts. Comparing it to $c^{\text{tlm-set(ovlp)}}$ shows the potential of allowing cuts to overlap. To avoid the prohibitive number of auxiliary variables, we consider projections to operator-counting variables: $c_{\text{all}}^{\text{dc-set(disj)}}$, $c_{\text{connected}}^{\text{dc-set(disj)}}$, $c_{\overline{\text{max}}}^{\text{dcset(disj)}}$, $c_{\text{atomic}}^{\text{dc-set(disj)}}$, and $c_{\text{atomic},\overline{\text{max}}}^{\text{dc-set(disj)}}$.

Table 1 shows how many problems we solve with the different operator-counting constraints. Reaching the time limit is the main reason for failure in these experiments, even for constraints with relatively few inequalities like $c_{\overline{\text{max}}}^{\text{dcset(disj)}}$.

We first observe that $c^{\text{tlm-set(disj)}}$ solves the fewest tasks among all tested configurations. Moreover, the ones it solves are generally solved the slowest. This confirms that the amount of variables in these constraints is too large. With $c^{\text{tlm-set(ovlp)}}$ we have the same number of variables and a comparable number of constraints but the heuristic quality is improved, leading to fewer state expansions. The cuts discovered with the overlapping variant are not guaranteed to be stronger than the ones by the disjoint variant. Theoretically the heuristic for $c^{\text{tlm-set(ovlp)}}$ does not dominate the one for $c^{\text{tlm-set(disj)}}$ but in practice, it always led to fewer expansions.

For disjoint cut sets, we have shown that $c_{\text{all}}^{\text{dc-set(disj)}}$ and $c_{\text{connected}}^{\text{dc-set(disj)}}$ are equivalent to $c^{\text{tlm-set(disj)}}$. Our experiments confirm this by yielding the same number of expanded states. Moreover, we find that projecting away auxiliary variables is beneficial, as $c_{\text{all}}^{\text{dc-set(disj)}}$ usually solves the problems significantly faster than $c^{\text{tlm-set(disj)}}$. The speed improvement is even more pronounced when considering $c_{\text{connected}}^{\text{dc-set(disj)}}$ which only includes the constraints for connected subsets of disj.

While the weaker constraints $c_{\overline{\text{max}}}^{\text{dcset(disj)}}$, $c_{\text{atomic}}^{\text{dc-set(disj)}}$ and $c_{\text{atomic},\overline{\text{max}}}^{\text{dc-set(disj)}}$ yield weaker heuristics, they still solve more problems than the other approaches. This is no surprise, as the number of connected subsets of disj can still be exponential in |disj|, while the number of inequalities for the weaker constraints is linear in |disj|. This speeds up the solver times for evaluating the heuristic in each state at the expense of heuristic quality. The difference in terms of coverage and planning time is marginal between these methods, with $c_{\overline{\text{max}}}^{\text{dcset(disj)}}$ coming out on top followed by $c_{\text{atomic}}^{\text{dc-set(disj)}}$. Even though these two are incomparable in theory, in practice $c_{\overline{\text{max}}}^{\text{dcset(disj)}}$ always expands fewer states than $c_{\text{atomic}}^{\text{dc-set(disj)}}$ suggesting that cases where $c_{\text{atomic}}^{\text{dc-set(disj)}}$ strengthens the LP are rare.

We also observe that $c_{\overline{\text{max}}}^{\text{dcset(disj)}}$ expands more states than $c_{\text{connected}}^{\text{dc-set(disj)}}$ in only 3 out of the 47 domains. It is hence a much cheaper relaxation that does not lose much information in practice. However, given that $c_{\text{atomic}}^{\text{dc-set(disj)}}$ results in comparable performance as $c_{\overline{\text{max}}}^{\text{dcset(disj)}}$, the benefit of transition counting over operator counting is limited for this particular set of cuts. We still believe the approach is worth pursuing further, as we have seen cases in our examples throughout the paper where transition counting adds information. Examples like the one in Figure 2 seem natural to us and not like rare exceptions. A direction worth exploring would be alternative cut generation methods to create more informed cuts.

## Conclusion

We introduced transition counting, an extension to the operator-counting framework. Transition-counting constraints can be formulated in any transition system but interesting planning tasks induce transition systems with a prohibitive amount of transitions. Our linking and translation constraints offer effective ways to formulate transition-counting constraints based on smaller abstractions with fewer transitions. They can then be used together with other operator-counting constraints for the full planning task.

Transition-counting constraints based on cuts can differentiate multiple uses of the same operator in different contexts. They dominate the landmark constraints derived from the same cuts, even if transition-counting variables are projected out. Projections to operator-counting variables such as $c_{\text{connected}}^{\text{dc-set}(\mathcal{C})}$ for pairwise disjoint cuts contain the same information as $c^{\text{tlm-set}(\mathcal{C})}$ but at a different size trade-off. Their relaxations give up some accuracy for a smaller size.

Experimentally, we have seen that just considering maximally connected subsets of $\mathcal{C}$ ($c_{\text{max}}^{\text{dc-set}(\mathcal{C})}$) had the best trade-off despite being an aggressive relaxation of $c_{\text{connected}}^{\text{dc-set}(\mathcal{C})}$. While we saw that distinguishing transitions in cuts can improve the heuristic value, this happened only rarely in practice and operator landmarks based on the same cuts perform almost as good. This might be an effect of our cut generation method. New ways of extracting cuts from abstractions, particularly ones that generate an interesting set of overlapping cuts, could lead to improved performance.

Another interesting line of future work is to add additional transition-counting constraints, such as for example (some relaxation of) state-equation constraints (van den Briel et al. 2007; Bonet 2013), or to consider larger abstractions.

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
