# OpenReview forum: "Transition Landmarks from Abstraction Cuts"
_icaps-conference.org/ICAPS/2024/Conference — ICAPS 2024_

### Official Review · Reviewer_f5zH · 2024-01-21

**Significance And Importance:** 2
**Soundness:** 4
**Novelty:** 3
**Clarity:** 3
**Overall Evaluation:** 2
**Confidence:** 4

**Weaknesses:**

1: Minor weaknesses that are easily fixable.

**Contributions Of The Paper:**

Tha paper focuses on operator-counting heuristics. It proposes an extension of
the heuristic by integrating landmarks extracted from cuts in abstract
transition systems.

**Ethical Considerations:**

(1) Not Applicable: The paper does not have any ethical considerations to address

**Nomination For Best Paper:**

No

**Questions For Authors:**

Feel free to address anything in my review. In particular, indicate whether
you're willing to fix the technical flaws I pointed out (or refute them if you
think I'm reading it wrong).

=== POST-REBUTTAL ===
Thank you for answers and clarifications.
Please update the paper for CR (assuming the paper is accepted) as you describe. In particular, regarding Def. 4 and the issue raised by the reviewer w1i6 regarding label reduction.

**Reproducibility:**

4: Authors promise to release code and domains (whichever apply).

**Strengths Of The Paper:**

The paper is very interesting and mostly easy to follow (see "Weaknesses" for
technical issues that need to be resolved). It mainly focuses on the integration
of cuts (landmarks) from abstract transition systems into the operator-counting
machinery which I think is important contribution in itself. However, it also
opens up a large space of possibilities of how to find these cuts, which kinds
of cuts play well together, and so on. I think this may lead to more interesting
research in the future.

**Weaknesses Of The Paper:**

I did not find any error in the reasoning, but there are some technical flaws
that need to be addressed for CR:

Inverse of label reduction \lambda (\lambda^{-1}) is not introduced (it is,
strictly speaking, not an inverse function, but rather mapping from labels to
sets of labels s.t. \lambda^{-1}(l) = {l' \mid \lambda(l') = l}).

First, it is stated that details of planning task encoding is not important
(first paragraph of Background), but later projections are discussed. The result
is that the definition of projections refers to variables that are not defined
at all. So, at least encoding of states should be introduced in the background.

The text should explicitly distinguish between a solution of the LP (i.e.,
constraints + the optimization criteria) and what is called "feasible solution"
throughout the text. At first I was confused and it was hard to follow the text
before I realized that what is meant by "feasible solution" is simply a variable
assignment satisfying all constraints. This really should be made prominent in
the text because the whole explanation is based on distinguishing the two -- in
the end, we minimize some criteria of the LP to get admissible estimates, but we
construct constraints so that it is satisfied by every plan.

The text does not distinguish between a constraint, set of constraints or set of
sets of constraints. The definition (Def. 1) says that a constraint is a set of
linear inequalities, but almost all propositions mix together constraints and
sets of constraints (and maybe even sets of sets -- I lost track). Please, fix the
notation in this regard, it is just a matter of choosing the correct notation.
For example, Prop. says {c,c'} is a constraint where c and c' are also
constraints -- just replace {c,c'} with c \cup c'. Or in other places there is
{c | ... } which can simply be replaced with \bigcup_{...}c and so on.

First sentence below Proposition 1:
What does it mean that a constraint "is in terms of the original operators"?

Sol_{IP} is never introduced, only Sol_{LP} is defined. Moreover, Sol_{LP} is
not a good definition, because it says Y and Z are variables, and then there is
Sol_{LP} = {Y \in R^n \mid \exists Z \in R^m ...} -- so, strictly speaking, Y
and Z suddenly become tuples of reals. Moreover, as per definition, Y and Z
become completely disconnected from the Y and Z from the assumption (the
definition says "Let .. Y ... Z" so Y and Z are fixed for Sol_LP(c), but
suddenly we use them to iterate over tuples of numbers). Please, fix this.
I wouldn't mind if you were less formal and simply state that Sol_{LP} is a set
of all variable assignments to operator-counting variables such that there
exists a variable assignment to auxiliary variables that together satisfy c (or
something along those lines). I would, however, object to using formalism that
simply is not correct.

Last paragraph of "Comparing Operator-Counting Constraints": It is not clear
what is meant by "can be extended to integer-valued solution". Extended how?
Isn't it simply that every integer-valued solution to LP is also a solution to IP?

Caption of Fig. 2 says that it shows TS with two cuts: TS has four cuts, but the
figure is explicitly marking just two -- please, just make the text more
accurate.

In several places, propositions say "X can be stronger/weaker". This is not very
accurate formal statement -- what is meant by it is that there exist a task with
X and Y in it such that X is stronger/weaker than Y. Either move these claims
out of propositions into text or make them more accurate mathematical statements. In
either case, please, explain better what exactly is meant by it because it is
not entirely clear (this is not just nit-picking, the propositions assume a
fixed task, so it might be confusing what these parts of the claims actually
mean).



Typos:
Background, Landmarks paragraph: "brevity.Let" --> missing space between "." and "Let"

page 5, left column in the middle: "two counts C_1,C_2 *are share* an operator"  --> remove "are"

---

> ### Author Rebuttal · Authors · 2024-01-25
>
> Thank you for your detailed feedback.
>
> (1) We will add a definition for the preimage operation. As you said, it is not
> the inverse because \lambda does not have to be bijective.
>
> (2) Thank you for pointing this out. The details of the encoding are not
> important for how we deal with transition cuts and the theoretical results. Our
> use case of deriving cuts from projections additionally requires assuming a
> factored representation of states. We will clarify this.
>
> (3) We will add a definition for "feasible solution".
>
> (4) We see a constraint as any condition that constrains the choices of values
> for the variables. In particular, a single linear inequality is a constraint,
> but a set of linear inequalities is a constraint as well. For example, requiring
> that three variables are equal is a constraint even though it takes multiple
> inequalities to represent it. In that sense a set of constraints is a
> constraint. But we see your point and will try to simplify the notation as you
> suggested.
>
> (5) In the setting of Proposition 1, constraint c uses operator-counting
> variables Y_o, where o is from O^\lambda. After the replacement, it will use
> operator-counting variables Y_o where o is from O.
>
> (6) Sol_{IP} is defined in line 215 as having the definition of Sol_{LP} but for
> integer-valued solutions rather than real-valued solutions.
>
> If the reviewers prefer, we can change Def. 4 to:
> Let c be an operator-counting constraint. The real-valued operator-counting
> solutions of c (denoted Sol_{LP}(c)) are real-valued assignments to the
> operator-counting variables such that an assignment to the auxiliary variables
> used in c exists where the combined assignment satisfies c.
>
> (7) We mean an extension in the sense of Def. 4. If constraint c uses
> auxiliary variables, an assignment to all operator-counting variables which
> we get from Sol_{LP} \cap N^n is enough to evaluate c.
> The paragraph above (lines 219-223) has an example where an assignment from
> Sol_{LP} \cap N^n cannot be extended to an integer solution.
>
> (8) We'll improve the caption of Fig. 2.
>
> (9) The statements in the proposition are relative to the quantified variables:
> a transition system T and a set of cuts C. The statement "X can be stronger than
> Y" refers to the possibility that for some T and some C, X is stronger than Y.
>
> We see this analogous to "Let x \in N_0 be a natural number and f(x) = 2x. Then
> f(x) is larger than or equal to x but can be equal to x." where "can" means
> "true for some choices of parameters".

---

### Official Review · Reviewer_PqpK · 2024-01-22

**Significance And Importance:** 2
**Soundness:** 3
**Novelty:** 3
**Clarity:** 3
**Overall Evaluation:** 1
**Confidence:** 3

**Weaknesses:**

1: Minor weaknesses that are easily fixable.

**Contributions Of The Paper:**

This work introduces a mechanism for leveraging transition counting in classical planning with abstractions.  The approach is analogous to operator counting but exploits dependencies of operators in transition systems, which can provide an advantage.  In fact, it is a generalization, because every transition-counting constraint is an operator-counting constraint by definition.

The authors show how abstraction cuts can be used to derive transition landmarks consisting of a disjunctive set of actions that must hold in every plan.  The resulting linear system can be stronger than the linear system obtained from operator landmarks.  Further characterizations (in terms of equivalence between cut-based constraints) are presented for cut sets that are disjoint, and for the cut sets that are overlapping.  Lastly, a method to find cuts in transition systems is presented.  Given the exponential space of possibilities, the authors aim to generate cuts that cover different parts of the transition system, with variants focusing on disjoint versus overlapping cuts.

The experimental evaluation on IPC instances (1998-2018) studies how different algorithmic choices influence the computation time.  For example, allowing the cuts to overlap results in more instances being solved.  The authors state, however, that "the benefit of transition counting over operator counting is limited for this particular set of cuts."

**Ethical Considerations:**

(1) Not Applicable: The paper does not have any ethical considerations to address

**Nomination For Best Paper:**

No

**Questions For Authors:**

- "To avoid the prohibitive number of auxiliary variables,
we consider projections to operator-counting variables" -> Can you clarify that these approaches are still based on the new transition cuts, i.e., the projection to operator counting variables can still be stronger than the traditional operator-only based cuts?

- A related question: Can you clarify that as a result the experimental evaluation does not contain a baseline comparison with operator-only based cuts?

- Can you give a bit more insight on the computational bottlenecks, in particular, as it relates to the LP/IP solving times?

**Reproducibility:**

4: Authors promise to release code and domains (whichever apply).

**Strengths Of The Paper:**

- The introduction of a new method, based on transition counting, for generating abstraction cuts.
- The theoretical characterization and comparison to existing related methods.  I found this the strongest part of the paper.
- The experimental evaluation: while it provides some insights, it was not clear to me how much value this new method brings w.r.t. the state of the art.

**Weaknesses Of The Paper:**

- The experimental comparison appears to be 'internal' only, as it compares the variants discussed in the paper.  Having an 'external' validation w.r.t. state of the art would be useful to provide evidence for the potential computational benefits.

---

> ### Author Rebuttal · Authors · 2024-01-25
>
> Thank you for your review! We would like to answer your first two questions together.
>
>
> > "To avoid the prohibitive number of auxiliary variables, we consider
> > projections to operator-counting variables"
> > -> Can you clarify that these approaches are still based on the new transition
> > cuts, i.e., the projection to operator counting variables can still be
> > stronger than the traditional operator-only based cuts?
> > A related question: Can you clarify that as a result the experimental
> > evaluation does not contain a baseline comparison with operator-only based
> > cuts?
>
> Yes, the approaches mentioned in that sentence are all based on transition cuts
> but do not use transition-counting variables. All of them dominate the cuts
> based on operator landmarks (c^{olm-set(C)}) and all except c^{dc-set(C)}_atomic
> can be stronger. This is because the constraint c^{dc-set(C)}_atomic is
> equivalent to c^{olm-set(C)} (see Figure 4). In fact, the c^{dc-set(C)}_atomic
> is not only equivalent to c^{olm-set(C)} but identical: it is the corner case of
> the general c^{dc-set(C)} family that leads to exactly the same linear
> inequalities as c^{olm-set(C)}. This can be seen in Figure 1: if S is limited to
> atomic sets S = {C}, then c^{dc(S)} = c^{olm(C)}.
>
> We thus see c^{dc-set(C)}_atomic as our baseline in the experiments.
>
> > Can you give a bit more insight on the computational bottlenecks, in
> > particular, as it relates to the LP/IP solving times?
>
> We considered the number of evaluated states per second. On average, we evaluate
> ~5400 states per second with our baseline c^{dc-set}_{atomic}. With the full
> transition landmarks (c^{tlm-set}) we only evaluate ~3600 states per second and
> assume the main cause for this reduction to be the significantly larger LP size
> which requires more solving time. When projecting to all connected components,
> we get back to ~5100 evaluations per second and restricting the set of
> considered connected components speeds it up further, but not quite reaching
> c^{dc-set}_{atomic}.

---

### Official Review · Reviewer_w1i6 · 2024-01-23

**Significance And Importance:** 2
**Soundness:** 3
**Novelty:** 3
**Clarity:** 4
**Overall Evaluation:** 2
**Confidence:** 4

**Weaknesses:**

2: No major or minor weaknesses.

**Contributions Of The Paper:**

This paper extends the operator-counting framework with additional constraints based on transition cuts in abstract state spaces. The resulting heuristics are analyzed both at a theoretical level, characterizing the informativeness and computational cost of diverse ways of expressing the cuts as constraints in the underlying LP.

**Ethical Considerations:**

(1) Not Applicable: The paper does not have any ethical considerations to address

**Nomination For Best Paper:**

No

**Questions For Authors:**

1) Is any of the configurations used equivalent to the operator-counting constraints without considering transition cuts?
2) Is any of the configurations used equivalent to the post-hoc optimization constraints without considering transition cuts?

**Reproducibility:**

4: Authors promise to release code and domains (whichever apply).

**Strengths Of The Paper:**

The paper has a relevant and significant contribution of introducing the notion of transition cuts in the operator-counting framework. Remarkably, the paper does not only present one way of encoding transition cut constraints, but rather analyzes multiple ways and compares them both theoretically and empirically. This makes for a very solid paper and a very good contribution to ICAPS.

**Weaknesses Of The Paper:**

The paper is well written and explained. However, there are a couple of things that I found confusing:

* The explanation of exact label reduction in the background without providing any reference, as this differs from the usual definition in the context of M&S abstractions. As far as I know, exact label reduction was introduced by Sievers et al. in "Generalized Label Reduction for Merge-and-Shrink Heuristics", AAAI'14. A thorough description is provided in the journal article "Merge-and-Shrink: A Compositional Theory of Transformations of Factored Transition Systems", JAIR 2021. As far as I know the term "exact label reduction" has not been used in other abstractions.
The key difference is that label reduction in the context of M&S may map operators to the same label if this does not add any additional transition, when considering the product with other transition systems. In the context of a single abstract transition system, all operators with the same cost could be reduced to the same label.

The definition in this paper is clear. But I'd use a different name or at least clarify that this differs from the definition used in M&S abstractions.

* In the experiments, I found it hard to understand what configurations correspond to previous work and which ones are new. If I understood correctly, the atomic variant is just the standard operator counting constraints, used in previous work.

* I am also missing some additional configurations from the related work, such as (saturated) post-hoc optimization and/or other work from the operator-counting framework (e.g. how far are the resulting heuristics from the nnOCP configuration in terms of expansions?)


Minor comments:
- Line 44: ‘planning task’ -> ‘planning tasks’.
- Line 69: ‘.Let’ -> ‘. Let’.
- Line 80: $\lambda^{-1}$ not defined.
- Line 265: $c^{tlm}$ constraints: $Y{t_1} + Y{t_2} \ge 1$ (2 instead of 3), $Y{t_3} + Y{t_4} \ge 1$ (3 instead of 2).
- Figure 3: why the disjoint variant does not find the dashed cut should be better explained.
- Line 354: two cuts C1, C2 are share -> remove are
- Line 493: missing final dot.
- Line 595: yields -> yield

---

> ### Author Rebuttal · Authors · 2024-01-25
>
> Thank you very much for your comments and kind review.
>
>
> (Q1)
> > Is any of the configurations used equivalent to the operator-counting
> > constraints without considering transition cuts?
>
> All our constraints are based on transition cuts but not all of them use
> transition-counting variables to talk about them. The constraint c^{olm-set(C)}
> in particular also ignores everything from the transition cuts except which
> operators are used. The constraint c^{dc-set(C)}_atomic is equivalent to it
> (even stronger, it is identical).
>
>
> (Q2)
> > Is any of the configurations used equivalent to the post-hoc optimization
> > constraints without considering transition cuts?
>
> We did not identify any variant of our heuristics that is equivalent to h^sPhO.
> The heuristic for c^{tlm-set}(C) dominates it for the right set of cuts but the
> dominance can be strict and hence they are not equivalent. The set of cuts
> needed for this dominance is the one we find with the "overlapping" variant of
> the backward-chaining method described in the paper. In the common case where
> all costs are 1, the cuts will be disjoint (even for the overlapping variant)
> and we can use the equivalence of c^{tlm-set(C)} to c^{dc-set(C)}_connected to
> get a version of the constraints without transition-counting variables that
> dominates sPhO.
>
>
> (Label reduction)
> > The explanation of exact label reduction [...] differs from the usual
> > definition
>
> You are right, our operation is not exactly the same as label reduction in
> merge-and-shrink where label reduction maintains a set of labels common to all
> abstractions. When reducing labels, merge-and-shrink does so globally, while our
> operation is local to one abstraction and can be seen as an efficient
> representation of its transitions. Rather than representing each transition
> separately, we only represent one in each equivalence class, reducing the number
> of {transition, operator}-counting variables.
>
> We used the term label reduction because our operation has the same effect when
> applied to an individual abstraction. We called it exact because the labels we
> combine subsume each other (in the notation of Sievers and Helmert) which would
> make the label reduction exact if the abstraction would be the only factor. But
> maybe the difference is too great and we should use our own term.
>
> We propose to rename "label reduction \lambda" to "label abstraction function
> \lambda" and not use the name "exact".

---

### Meta-Review · Area_Chair_h44i · 2024-02-04

**Recommendation:** Accept (Oral)
**Confidence:** 5

**Metareview:**

The authors introduce the notion of transition-counting constraints for the operator counting framework. Computing them exactly can be prohibitively expensive, thus the authors devise approximation methods and compare them.

This is a strong paper that is well suited for ICAPS.

**Ethical Considerations:**

(1) Not Applicable: The paper does not have any ethical considerations to address